# INDUCTIVE VISUAL LOGIC FOR FEW-SHOT OUT-OF-DISTRIBUTION ADAPTATION IN VLMS

## ABSTRACT

Few-shot visual reasoning requires models not only to learn from limited supervision while also adapting across domains, including those that are far from pretraining distributions. Modern vision-language models (VLMs) such as Qwen and LLaVA excel in zero-shot tasks while collapsing in these distant out-of-distribution (OOD) settings, where standard adaptation methods provide limited gains. We introduce **I**nductive **V**isual **L**ogic (IVL), a trait-based reasoning framework that extracts visual traits through dual-mode prompting (semantic and low-level features) and organizes them into compact, interpretable dictionaries. IVL applies inductive–deductive reasoning over these traits at inference and grounds predictions in transferable explanations without updating model weights. Through reasoning over traits rather than memorizing examples, IVL enables training-free few-shot adaptation that explicitly addresses both near-domain shifts and distant OOD shifts. Our experiments across multiple datasets demonstrate that IVL improves few-shot performance while producing more interpretable predictions. Our evaluation results and insights highlight trait-level reasoning as a scalable and complementary path toward robust OOD adaptation in foundation-scale VLMs.

## 1 INTRODUCTION

Rapid adaptation from minimal examples has become increasingly essential for deploying vision-language models (VLMs) in diverse real-world applications. Critical domains cannot collect large datasets for traditional training approaches, as medical imaging faces stringent privacy regulations, industrial inspection involves proprietary constraints, and scientific research encounters phenomena with inherently limited sample availability. Current VLMs demonstrate substantial limitations when confronted with genuinely novel visual taxonomies absent from their pretraining data, including rare pathologies, specialized manufacturing defects, or domain-specific categorization systems. When VLMs encounter distant out-of-distribution (OOD) concepts, they lack the relevant visual primitives acquired during pretraining to effectively distinguish between classes. Gradient-based fine-tuning approaches suffer from overfitting to spurious patterns when provided with limited examples (Zhang et al., 2020; Geirhos et al., 2020), while in-context learning (ICL) only reweights existing but fundamentally inadequate representations (Min et al., 2022). This fundamental limitation prevents models from being effectively applied to OOD data with minimal pretraining exposure. As demonstrated by Li & Flanigan (2024), foundation models often achieve strong performance on "few-shot" tasks primarily since they have already encountered these specific task distributions during pretraining, while exhibiting dramatic performance degradation on truly novel task categories.

Motivated by this challenge, this work investigates a specific class of adaptation problems: **distant out-of-distribution (distant-OOD)** scenarios where target domains share minimal overlap with pretraining data, as opposed to near-OOD cases that retain substantial distributional similarity to training corpora. **Distant-OOD** tasks exhibit a characteristic failure pattern that differs fundamentally from conventional few-shot scenarios: they demonstrate poor zero-shot performance with negligible or negative gains from few-shot adaptation. Unlike near-OOD tasks where fine-tuning provides substantial improvements, distant-OOD scenarios show minimal enhancement or performance degradation across existing adaptation methods. This phenomenon occurs consistently across two evaluation frameworks: real-world datasets where adaptation fails, including specialized medical imaging, rare species identification, and industrial defect detection, as well as a controlled synthetic benchmark specifically designed for this analysis (Section 4.2.2). For instance, when adapting vision-language

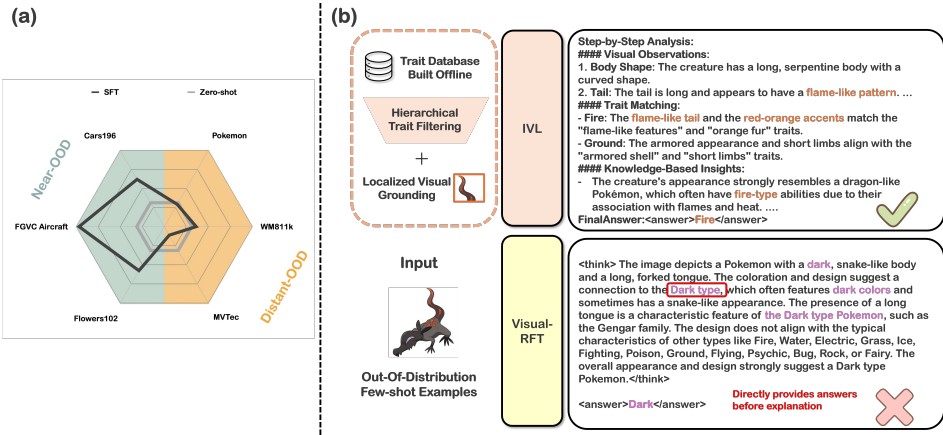

Figure 1: (a) Distant-OOD vs. near-OOD performance. The radar diagram illustrates how fine-tuning methods perform across different datasets. (b) IVL addresses the efficiency challenge more effectively than gradient-based methods such as SFT and Visual-RFT. The output logic comparison demonstrates that Visual-RFT directly provides answers before explanation, which deviates from human reasoning patterns. In contrast, IVL first analyzes observed features, compares them with the traits dictionary, refines them with prior knowledge, and then proposes the final decision. This logic aligns more closely with human cognitive processes and therefore enhances model interpretability.

models to identify rare pathological conditions or novel manufacturing defects, standard methods demonstrate the same consistent failure patterns. Across all experimental settings, supervised fine-tuning (SFT), LoRA (Hu et al., 2022), Visual-RFT (Liu et al., 2025), and in-context learning (ICL) demonstrate consistent failure patterns (Fig. 1). This systematic failure pattern reveals the limitation that without relevant visual primitives from pretraining, gradient updates simply overfit to spurious correlations in the few available examples, while ICL cannot activate knowledge that was never acquired during pretraining. These findings suggest that distant-OOD adaptation requires fundamentally different approaches that can discover and compose visual patterns from limited examples without relying on absent pretraining features. Rather than adapting parameters within inadequate representation spaces, effective solutions must construct new visual distinctions from minimal data.

In light of these failures, this study proposes **I**nductive **V**isual **L**ogic (IVL), a framework that addresses distant-OOD challenges through explicit trait-based reasoning rather than implicit parameter adaptation. The key insight underlying IVL stems from cognitive science research on human visual learning: when humans encounter unfamiliar categories, they systematically identify salient patterns across limited examples and document which visual attributes such as colors, shapes, textures, or structural components appear consistently within each category (Lake et al., 2015; Tenenbaum et al., 2011). Contemporary VLMs fundamentally lack this capacity for systematic visual pattern discovery and rely instead on implicit feature representations that may be entirely absent for distant-OOD domains. IVL operationalizes human-like visual reasoning through a dual-mode trait extraction process that captures both semantic knowledge and primitive visual features from support examples, clusters semantically similar traits into canonical descriptors, and constructs explicit classification rules for inference-time application. Through explicit construction of classification rules from observable visual traits, IVL circumvents the representational limitations that cause gradient-based methodologies to fail on distant-OOD tasks. Our main contributions can be summarized as follows:

- **Distant-OOD characterization:** We provide systematic analysis of the underlying mechanisms through which VLMs fail catastrophically on truly novel visual concepts and establish protocols that distinguish distant-OOD from near-OOD regimes while showing that representational absence rather than distributional shift constitutes the fundamental barrier.
- **Trait-based reasoning framework:** We introduce a training-free methodology that constructs interpretable classification rules through visual trait extraction and organization, enabling effective adaptation where parametric methods fail due to representational gaps.
- **Cognitive-computational bridge:** We show how insights from human visual learning can be operationalized for VLM adaptation and establish trait-based reasoning as a complementary paradigm to gradient-based optimization for deployment in specialized domains.

## 2 RELATED WORK

### 2.1 GENERATIVE VISION-LANGUAGE MODELS

Generative VLMs (Yin et al., 2024) such as LLaVA (Liu et al., 2023) and Qwen2.5-VL (Bai et al., 2025) have demonstrated remarkable zero-shot (Dai et al., 2023) and chain-of-thought (CoT) reasoning (Zhang et al., 2023) capabilities through large language model (LLM) backbones that generate textual responses conditioned on visual inputs. These models, typically pre-trained on massive web-scale data, can achieve impressive performance on standard benchmarks and diverse vision-language tasks without task-specific fine-tuning. However, despite their broad knowledge foundation, generative VLMs typically exhibit significant performance degradation when encountering substantial domain shifts. When deployed on specialized domains, these models struggle due to their reliance on high-level semantic patterns from natural images, which fail to transfer when low-level visual statistics differ significantly (Li et al., 2023). Moreover, adapting these VLM models presents unique challenges: their billion-scale parameters make full fine-tuning process computationally prohibitive, while parameter-efficient methods like LoRA (Hu et al., 2022) demonstrate limited effectiveness when domain gaps are large. These computational and performance challenges have motivated researchers to develop various few-shot adaptation strategies that are specifically tailored for VLMs.

### 2.2 FEW-SHOT DOMAIN ADAPTATION WITH VLM

Given the computational constraints and domain shift challenges, few-shot domain adaptation has emerged as a practical paradigm for deploying VLMs in specialized domains with limited labeled data. Existing methods can be broadly categorized into three major approaches, each attempting to balance adaptation effectiveness with computational efficiency. Prompt learning methods, such as CoOp (Zhou et al., 2022b) and CoCoOp (Zhou et al., 2022a), learn continuous prompts that adapt to novel domains without modifying the backbone model. CasPL (Wu et al., 2024) further enhances domain generalization through cascade training combined with pseudo-labeling, while Nemesis (Fu et al., 2024) improves stability through normalization of soft prompt vectors. In addition to prompt-based approaches, parameter-efficient fine-tuning (PEFT) methods like LoRA, MMA (Yang et al., 2024) and PACE (Ni et al., 2024) insert lightweight adapters or apply consistency regularization to adapt to new domains with minimal computational overhead. Recently, reinforcement learning (RL) has emerged as another promising direction for VLM adaptation. Visual-RFT (Liu et al., 2025) and Chu et al. (2025) showed that RL-guided fine-tuning enables models to rapidly adapt to new domains while maintaining generalizability to related concepts. Nevertheless, recent studies (Li & Flanigan, 2024) and our experiments reveal these methods fail to improve on distant-OOD tasks. While prompt tuning assumes compositional recombination of existing features (Zhou et al., 2022b), and PEFT methods rely on feature adaptation (Hu et al., 2022), neither can handle cases where discriminative visual primitives are absent. Even Visual-RFT (Liu et al., 2025), which uses RL for adaptation, performs worse than simple supervised fine-tuning on distant-OOD tasks, suggesting that RL's exploration cannot discover features that fundamentally do not exist in the model's representation space. This universal failure across gradient-based methods, prompt tuning, and RL suggests that distant-OOD needs fundamentally different approaches beyond parameter adaptation.

### 2.3 COMPOSITIONAL REASONING LIMITATIONS IN VLMS

Recent work reveals compositional failures in VLMs that directly explain distant-OOD adaptation challenges. Thrush et al. (2022) showed that state-of-the-art VLMs achieve near-random performance on novel compositions of known concepts, while Ma et al. (2023) revealed systematic failures in compositional generalization despite recognition of individual components. Yuksekgonul et al. (2022) identified that VLMs rely on bag-of-words representations rather than true compositional understanding, which explains their brittleness on out-of-distribution compositions. These limitations are critical for distant-OOD tasks, which require composition of new visual representations from primitive features precisely where VLMs fail. When encountering novel categories, models cannot systematically combine visual attributes into coherent concepts, which causes gradient-based adaptation to fail regardless of optimization strategy. IVL addresses this limitation through complete bypass of internal composition mechanisms. Instead of reliance on the model's flawed compositional reasoning, we explicitly extract visual traits through structured prompting and combine them into interpretable classification rules, enabling effective adaptation where parametric methods fail.

## 2.4 TRAIT-BASED AND NEURO-SYMBOLIC APPROACHES

Attribute-based recognition has long explored interpretable alternatives to end-to-end learning. Classical work on visual attributes (Farhadi et al., 2009; Lampert et al., 2009) demonstrated that modeling explicit visual properties enables zero-shot recognition through attribute composition. Recent concept bottleneck models (Koh et al., 2020) enforce interpretability through prediction of human-understandable concepts before classification, though requiring extensive concept annotations. In the VLM era, several works explore trait-like representations. Menon & Vondrick (2022) used GPT-3 to generate visual descriptors for classification while relying solely on the model's semantic knowledge. Pratt et al. (2023) showed that VLM-generated descriptions can improve CLIP's zero-shot performance while not addressing few-shot learning from novel domains. Unlike these approaches, IVL leverages pretrained VLM capabilities through dual-mode prompting to extract both semantic knowledge and low-level visual features from few-shot examples. This dual extraction strategy proves crucial for distant-OOD where semantic knowledge alone fails, as the model can still describe primitive visual features even when high-level semantic understanding is absent.

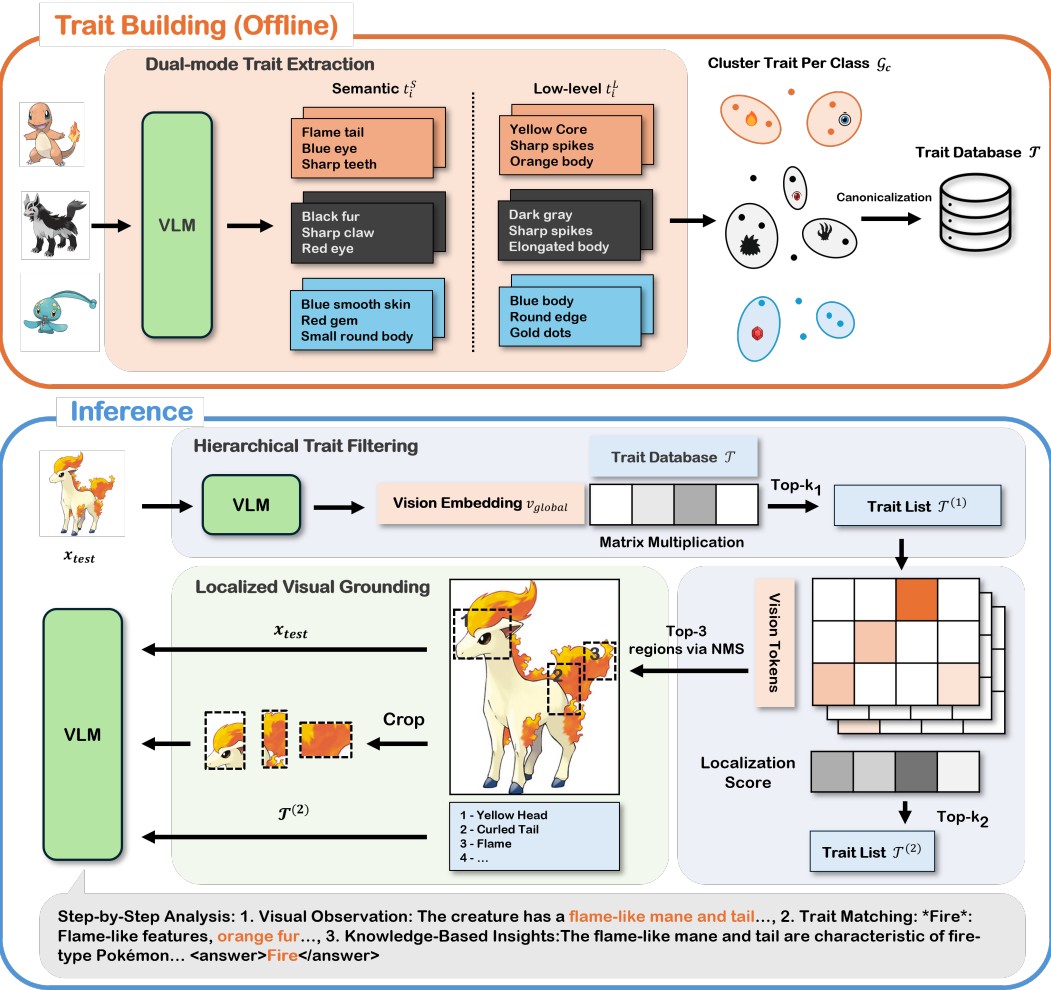

Figure 2: Framework Overview. Offline Trait Building: The system extracts semantic ($t_i^S$) and low-level ($t_i^L$) traits from support images using a VLM, clusters them per class, assigns canonical names, and stores them in database $\mathcal{T}$. Inference: Given test image $x_{\text{test}}$, hierarchical filtering is performed: (1) compute $\mathbf{s} = \mathbf{E}_{\mathcal{T}}^T \mathbf{v}_{\text{global}}$ to select top-$k_1$ traits $\mathcal{T}^{(1)}$, (2) compute patch-trait attention scores for localized grounding while selecting top-$k_2$ traits $\mathcal{T}^{(2)}$ and extracting top-3 regions via NMS, and (3) prompt the VLM with the original image, cropped regions, and derive $\mathcal{T}^{(2)}$ for final classification.

## 3 METHODOLOGY

### 3.1 PROBLEM FORMULATION

In light of the above limitations, this section first formulates the distant-OOD adaptation problem within a trait-based reasoning framework. IVL addresses the $n$-shot $C$-way classification problem: given a support set $\mathcal{D} = \{(x_i, y_i)\}_{i=1}^{n \times C}$ with $n$ examples per class and $C$ classes, the objective is to classify query image $x_q$ into one of these $C$ classes. The key insight underlying IVL involves transforming few-shot classification from direct image matching to explicit trait-based reasoning. A *trait* $t$ is defined as a textual description of a visual feature (e.g., "red wings", "spotted pattern", etc.). From the support set, IVL constructs a trait database $\mathcal{T}$ where each class $c$ is represented by a set of traits $\mathcal{T}_c = \{t_1^c, t_2^c, ..., t_{m_c}^c\}$ extracted from its support images. The full database $\mathcal{T} = \bigcup_{c=1}^{C} \mathcal{T}_c$ can contain thousands of traits. This explicit trait representation enables classification decisions grounded in interpretable visual clues rather than relying on potentially absent pretrained features.

### 3.2 OVERVIEW

IVL addresses distant-OOD challenges through a two-stage architecture that explicitly constructs visual knowledge rather than adapting existing representations. Fig. 2 illustrates the two-operation stages of IVL: (1) an offline trait building stage that systematically extracts and organizes discriminative patterns from the support set into $\mathcal{T}$, and (2) an inference stage that efficiently retrieves relevant traits for classification. This two-stage design enables interpretable decisions through explicit trait matching while maintaining manageable computational requirements through hierarchical filtering. During the **Trait Building Stage (Offline)**, IVL extracts discriminative visual traits through dual-mode prompting, clusters semantically similar traits, and applies quality filtering to construct $\mathcal{T}$. The **Inference Stage** employs hierarchical filtering to reduce thousands of traits to approximately $100 \sim 200$ most relevant ones depends on the size of dataset: first through embedding similarity, then cross-modal attention verification, and finally through extraction of salient image regions for focused classification. This ensures that classification decisions are grounded in explicit visual evidence rather than implicit feature representations that may be absent in distant-OOD scenarios.

### 3.3 TRAIT BUILDING STAGE

This stage constructs a structured trait vocabulary from raw visual observations through a three-phase pipeline: dual-mode trait extraction, semantic clustering, and canonical naming. Given a few-shot training dataset $\mathcal{D}$ with $n$ examples per class across $C$ classes, where each example consists of an image $x_i$ and its class label $y_i \in \{1, ..., C\}$, this process constructs a trait database $\mathcal{T}$ that maps each class to discriminative visual descriptors. The construction transforms raw VLM-generated descriptions into a refined vocabulary of canonical traits that are suitable for inference-time reasoning.

#### 3.3.1 DUAL-MODE TRAIT EXTRACTION

The trait extraction process addresses a fundamental limitation in VLMs when processing distant-OOD data: semantic prompting frequently triggers *description collapse*, which produce generic category labels rather than discriminative visual features. For each image $x_i$, two complementary trait types are extracted to ensure robust feature representation over varying degrees of domain shift:

**Semantic traits** $t_i^S$: Extracted via prompt $p_S$ that explicitly requests knowledge-based descriptions (see Appendix D), leveraging the model's pre-trained understanding when applicable. These traits excel in familiar domains where the model can provide rich categorical and functional attributes.

**Low-level traits** $t_i^L$: Extracted via prompt $p_L$ that constrains responses to primitive visual features (e.g., colors, shapes, textures, spatial relations, etc.), and explicitly avoids category labels. These traits aim to maintain descriptive richness even for novel domains where semantic understanding fails, for instance, describing "white wrinkled regions in center" rather than simply "retinal tissue".

This dual extraction strategy builds upon self-consistency principles (Wang et al., 2022) while adapting the approach from filtering to enrichment. Rather than seeking consensus, the process performs $K$ independent extractions per image (typically $K = 5$), with each extraction generating $5 \sim 10$

traits. All unique trait variants are then preserved across extractions, including lexical variations such as "crimson", "red", and "reddish", which are subsequently consolidated through clustering.

### 3.3.2 TRAIT CLUSTERING AND CANONICALIZATION

The trait extraction process produces extensive and redundant trait collections requiring systematic consolidation. For class $c$, let $\mathcal{T}_c = \bigcup_{i:y_i=c} t_i$ represent all extracted traits. The consolidation process begins with preprocessing that normalizes traits through lowercase conversion, punctuation removal, and artifact filtering, followed by frequency counting $f(t, c)$ for each trait $t$ in class $c$. To integrate semantically similar traits, IVL uses Sentence Transformers (Reimers & Gurevych, 2019) to embed unique traits as $\mathbf{e}_t \in \mathbb{R}^{d_{\text{sent}}}$ followed by HDBSCAN (McInnes et al., 2017) clustering as:

$$\mathcal{G}_c = \text{HDBSCAN}(\{\mathbf{e}_t : t \in \mathcal{T}_c\}), \tag{1}$$

where $\mathcal{G}_c = \{G_1^c, G_2^c, ...\}$ represents the trait clusters for class $c$, with HDBSCAN parameters `min_cluster_size=2`, `min_samples=1` to preserve rare but potentially discriminative traits.

IVL specifically selects Sentence Transformers over VLM text encoders based on empirical observations: VLM encoders, optimized for vision-text alignment, over-compress textual semantic spaces, typically producing only 2-3 clusters per class regardless of trait diversity. This compression collapses crucial distinctions. Sentence Transformers, optimized for semantic similarity, maintain appropriate granularity while grouping genuine synonyms. Rather than using geometric centroids, each cluster $G_j^c$ generates representative descriptors through VLM compositional reasoning. The VLM synthesizes cluster members into interpretable canonical names, transforming collections such as {"red legs", "crimson arms", "reddish hands"} into unified descriptors like "red limbs". This produces semantically coherent representations maintaining interpretability while reducing redundancy.

### 3.3.3 TRAIT REFINEMENT AND QUALITY CONTROL

The canonicalized traits undergo type-aware filtering to ensure quality while preserving discriminative information. The quality control strategy adapts to trait characteristics: **Semantic traits** $t_i^S$ undergo relevance filtering using VLM common-sense reasoning to remove obviously inconsistent traits (e.g., "aquatic features" for bird classes). **Low-level traits** $t_i^L$ skip relevance filtering, as VLMs cannot reliably assess the importance of primitive features for novel domains where semantic understanding may be absent. Following type-specific filtering, semantic and low-level traits are merged to form the complete trait database $\mathcal{T}$. To enable efficient retrieval during inference, embeddings are pre-computed for all retained traits, establishing the foundation for the subsequent inference stage.

### 3.3.4 TRAIT EMBEDDING DATABASE CONSTRUCTION

The full trait database can contain up to $10,000$ traits depending on dataset complexity. For inference efficiency, embeddings are pre-computed for each trait using Qwen2.5-VL's text encoder rather than the Sentence Transformers used for clustering. This encoder selection proves essential since inference requires comparing visual features against text traits in a shared representation space, which only VLM encoders provide through their joint vision-text training. Specifically, each trait description $t_i \in \mathcal{T}$ is encoded to obtain its embedding $\mathbf{e}_i \in \mathbb{R}^d$, where $d$ denotes the dimensionality of Qwen2.5-VL's joint embedding space. These embeddings are concatenated to form the matrix $\mathbf{E}_{\mathcal{T}} = [\mathbf{e}_1, \mathbf{e}_2, ..., \mathbf{e}_{|\mathcal{T}|}] \in \mathbb{R}^{d \times |\mathcal{T}|}$, which enables efficient trait retrieval via matrix operations during inference and completes the trait building stage preparation for the downstream classification tasks.

## 3.4 INFERENCE STAGE

Given the trait database $\mathcal{T}$ constructed in Section 3.3, this stage enables efficient few-shot classification through hierarchical trait filtering and localized visual grounding for distant-OOD scenarios.

### 3.4.1 GLOBAL TRAIT RETRIEVAL

Given a test image $x_{\text{test}}$, IVL first extracts its visual representation using the Qwen2.5-VL vision encoder. The vision transformer processes the image into $P$ patch tokens $\{v_1, v_2, ..., v_P\}$, where

each $v_i \in \mathbb{R}^d$ is a spatial region. A global image embedding is computed through mean pooling:

$$\mathbf{v}_{\text{global}} = \frac{1}{P} \sum_{i=1}^{P} v_i \in \mathbb{R}^d. \tag{2}$$

To retrieve relevant traits, cosine similarities are computed between the global visual embedding and all trait embeddings through normalized matrix multiplication: $\mathbf{s} = \hat{\mathbf{E}}_{\mathcal{T}}^T \hat{\mathbf{v}}_{\text{global}} \in \mathbb{R}^{|\mathcal{T}|}$, where $\hat{\mathbf{E}}_{\mathcal{T}}$ is the L2-normalized trait embedding matrix and $\hat{\mathbf{v}}_{\text{global}}$ is the L2-normalized global visual embedding. The joint vision-text training of the encoder ensures that both modalities reside in a unified $d$-dimensional representation space, enabling similarity computation without additional projection layers. Based on $\mathbf{s}$, the top-$k_1$ traits are selected to establish the globally-filtered trait set $\mathcal{T}^{(1)} \subset \mathcal{T}$.

### 3.4.2 LOCALIZED TRAIT REFINEMENT

While global retrieval reduces the search space, identifying traits with strong visual grounding in the test image requires further refinement. Following Kang et al. (2025), IVL leverages internal attention mechanisms of the VLM to measure trait-image alignment. For each trait $t \in \mathcal{T}^{(1)}$, both the test image and trait text are fed to the VLM, and attention maps are extracted from pre-identified grounding heads (calculated on RefCOCO dataset). These attention maps reveal which image patches the model associates with each trait. A localization score is computed by taking the maximum attention weight across all image patches, aggregated over the top-3 grounding heads. This score measures whether the trait has strong visual evidence in any specific part of the image. The top-$k_2$ traits with highest localization scores are selected to form $\mathcal{T}^{(2)} \subset \mathcal{T}^{(1)}$. Unlike the initial global filtering which relies on overall similarity, this refinement ensures retention of traits that have specific visual grounding in the test image, with their attention maps indicating where to focus visual analysis.

### 3.4.3 VISUAL REGION EXTRACTION

The cross-modal attention maps provide spatial localization for the selected traits. To extract discriminative regions, attention weights are converted to 2D spatial heatmaps and smoothed using Gaussian filtering to identify coherent regions of interest. Local maxima are then identified within these smoothed attention maps, and bounding boxes are extracted around the top-3 peaks with highest cumulative attention across all selected traits in $\mathcal{T}^{(2)}$. These extracted regions concentrate visual analysis on the most trait-relevant areas, offering focused visual context for the final classification.

### 3.4.4 CLASSIFICATION WITH MULTI-MODAL CONTEXT

With both discriminative traits and visual regions identified, the final classification stage integrates this multi-modal evidence for decision making. Qwen2.5-VL is prompted with three complementary inputs: the original test image $x_{\text{test}}$, the refined trait list $\mathcal{T}^{(2)}$ as textual context describing discriminative features, and the three extracted cropped regions for focused visual inspection. This hierarchical approach systematically combines three levels of visual understanding to produce the final classification. Global similarity filtering identifies broadly relevant traits through matrix multiplication operations, localized attention refinement grounds these traits spatially through cross-modal attention mechanisms, and focused regional analysis concentrates on the most discriminative image areas through cropped region inspection. The resulting multi-modal reasoning process mimics human visual classification strategies that integrate holistic scene understanding and detailed feature analysis.

## 4 EXPERIMENTAL RESULTS

### 4.1 EXPERIMENT SETUP

The proposed framework is evaluated against established baselines across a carefully curated suite of out-of-distribution tasks to validate its effectiveness on distant-OOD scenarios. Consistent experimental conditions are maintained across all methods to ensure reproducibility and fair comparison.

**Model Architecture and Baselines.** Qwen2.5-VL-7B serves as the base VLM for all VLM-based approaches, including the proposed IVL method and comparison baselines: zero-shot inference, supervised fine-tuning (SFT), LoRA with SFT, Visual-RFT (Liu et al., 2025), and standard in-context

learning (ICL). For the CLIP baseline, the ViT-B/32 architecture is utilized. The controlled comparison isolates the contribution of our trait-based learning paradigm from architectural confounds.

**Few-Shot Sampling Protocol.** For each task, a support set $\mathcal{D} = \{(x_i, y_i)\}_{i=1}^{n \times C}$ is sampled from the full training data, where each class $c \in \{1, ..., C\}$ contributes exactly $n$ examples for $n$-shot learning. To eliminate sampling variance as a confounding factor, deterministic sampling with a fixed random seed is employed across all experiments. This rigorous experimental design ensures that performance differences reflect the inherent capabilities of each method rather than statistical artifacts, providing a robust evaluation of trait-based reasoning for distant-OOD adaptation challenges.

## 4.2 OUT-OF-DISTRIBUTION DATASETS

### 4.2.1 PUBLIC OOD DATASET

A critical limitation of existing few-shot adaptation benchmarks for VLMs lies in their insufficient distributional shift from pretraining data. Previous work (Liu et al., 2025) demonstrates that standard OOD benchmarks can achieve near-ceiling performance with minimal fine-tuning, which suggests substantial task-relevant knowledge already encoded during pretraining. This phenomenon undermines the validity of these benchmarks for assessment of true few-shot generalization capabilities.

In this study, the concept of **distant out-of-distribution (distant-OOD)** tasks is introduced to address this limitation, and is empirically identified as tasks where the adaptation gain $\Delta = \text{Acc}_{\text{few-shot}} - \text{Acc}_{\text{zero-shot}}$ remains below a threshold $\tau = 0.15$ even after few-shot fine-tuning. This threshold emerged from natural clustering in preliminary experiments across 22 datasets (treat each MVTech AD (Bergmann et al., 2021) category as a individual dataset) , where tasks separated into two distinct groups with minimal overlap around this boundary (see Appendix B for detailed analysis). Based on this criterion, several vision classification datasets from public repositories are systematically evaluated, primarily sourced from Kaggle competitions and specialized computer vision benchmarks. Table 3 presents the selected distant-OOD datasets that satisfy the established criteria and exhibit persistent adaptation challenges despite few-shot training. These datasets span medical imaging (Retinal OCT (Naren, 2021)), industrial inspection (WM811k (WM811K, 2023)), and fine-grained classification tasks where domain-specific visual patterns were absent from pretraining data. The selection of these benchmarks ensures evaluation of adaptation methods under truly novel visual scenarios where prior approaches consistently fail to improve upon zero-shot performance.

### 4.2.2 POKEMON DATASET

To investigate adaptation dynamics under controlled conditions, a synthetic challenging distant-OOD benchmark is constructed based on a task "Pokémon type classification". Given an image $x \in \mathcal{X}$ depicting a Pokémon creature, the task requires prediction of its elemental type $y \in \{\text{Fire, Water, Grass, Electric, Psychic, Fairy, ...}\}$ based solely on visual attributes. This synthetic benchmark provides three methodological advantages for systematic evaluation. First, it establishes systematic visual-semantic mappings that combine simple associations (red/orange coloration indicating Fire type) with complex patterns (subtle features distinguishing Psychic from Fairy types), which enables decomposition of adaptation performance over different reasoning difficulties. Second, it enables human baseline comparison since children easily learn these visual-type associations through brief exposure, highlighting the gap between human and VLM visual reasoning capabilities. Third, it offers controlled evaluation with known ground-truth discriminative features, enabling verification of whether methods learn intended patterns or exploit spurious correlations.

The Pokémon dataset complements real-world distant-OOD tasks through provision of a unique test case: while children quickly learn visual-type associations through brief gameplay, VLMs consistently fail to improve with fine-tuning. This paradox, where a task trivial for humans remains intractable for VLMs despite fine-tuning, helps isolate whether adaptation failures stem from optimization difficulties or fundamental representation gaps. The systematic visual design provides ground truth for evaluation of which features methods actually learn versus spurious correlations they exploit. This controlled benchmark thus serves as a critical diagnostic tool for understanding the limitations of current adaptation approaches when faced with truly novel visual reasoning tasks.

Table 1: Experimental Results. Comparing n-shot performance by methods.

| Model | Method | n-shot | Pokemon | Retinal OCT | WM811k | MVTec AD* |
|-------|--------|--------|---------|-------------|--------|-----------|
| Qwen2.5-VL | None | 0 | 48.92 | 18.75 | 11.45 | 34.88 |
| | SFT | 1 | 47.25 | 0.00 | 12.41 | 39.46 |
| | | 8 | 52.85 | 33.39 | 15.90 | – |
| | SFT+LoRA | 1 | 49.15 | 13.69 | 9.16 | 37.16 |
| | | 8 | 48.47 | 14.00 | 9.16 | – |
| | RFT | 1 | 48.47 | 14.00 | 9.52 | 36.71 |
| | | 8 | 49.72 | 13.14 | 8.92 | – |
| | ICL | 1 | 22.0 | 13.75 | 12.41 | 24.45 |
| | | 8 | 34.24 | 19.54 | 14.22 | – |
| | **IVL (Ours)** | 1 | **51.00** | **20.83** | **13.53** | **40.69** |
| | | 8 | **56.30** | **25.75** | **16.70** | – |
| **ViT-B/32** | CLIP | 0 | 42.25 | 12.29 | 13.49 | 39.12 |

*Detailed description can be found at Appendix C.

## 4.3 QUANTITATIVE RESULTS

Table 1 presents a comprehensive evaluation of various adaptation methods across four distant out-of-distribution (OOD) datasets, spanning medical imaging, industrial inspection, and natural image classification domains. Performance was assessed under both 1-shot and 8-shot configurations (with MVTec AD detailed in Appendix C) to evaluate computational efficiency across different data availability scenarios. The proposed IVL method achieves superior overall performance with consistent improvements across all evaluated datasets, demonstrating computational efficiency through training-free adaptation. In the 1-shot configuration, IVL outperforms zero-shot baselines by 2.08% on Pokemon, 2.08% on Retinal OCT, 2.08% on WM811k, and 5.81% on MVTec AD. Performance gains become more pronounced with additional examples, reaching 56.30% on Pokemon in the 8-shot configuration (+7.38% over baseline), indicating efficient utilization of available training examples. Gradient-based methods exhibit substantial performance variance across domains, highlighting computational inefficiency through inconsistent adaptation. SFT demonstrates catastrophic failure on Retinal OCT in 1-shot scenarios (0.00%) despite strong performance on MVTec AD (39.46%), suggesting severe overfitting in low-data medical imaging scenarios that wastes computational resources. In-context learning (ICL) underperforms across all datasets (e.g., 22.00% on Pokemon versus 48.92% baseline), indicating that naive visual example concatenation proves computationally inefficient for complex domain adaptation tasks. These empirical results clearly demonstrate that IVL successfully addresses the computational inefficiencies of gradient-based methods through overfitting mitigation while providing more robust and efficient adaptation than context-based approaches. The consistent performance improvements across diverse domains validate the computational advantages of the proposed trait-based reasoning frameworks for distant-OOD scenarios.

## 5 CONCLUSION

This paper introduced IVL, a training-free framework that addressed VLM failures on distant-OOD tasks through trait-based reasoning rather than parameter adaptation. The key insight that VLMs failed due to absent visual primitives motivated development of explicit classification rules from observable features instead of adapting inadequate pretrained representations. IVL combined semantic understanding with primitive visual features through dual-mode extraction and hierarchical filtering to identify discriminative traits while preserving interpretability. Our experiments validated substantial improvements on distant-OOD benchmarks where gradient-based methods failed, showing that parameter adaptation cannot overcome fundamental representation gaps and establishing trait-based reasoning as essential for VLM deployment in specialized domains distant from pretraining data.

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

## A NOTATION TABLE

Table 2: Summary of Notations used in IVL

| Notation | Description |
|---|---|
| $t$ | Trait |
| $t^S$ | Semantic traits |
| $t^L$ | Low-level traits |
| $\mathcal{T}$ | Trait database |
| $\mathcal{T}_c$ | Traits extracted from class $c$ |
| $\mathcal{T}^{(1)}$ | Globally-filtered trait set (stage 1) |
| $\mathcal{T}^{(2)}$ | Locally-refined trait set (stage 2) |
| $\mathcal{G}_c$ | Set of trait clusters for class $c$ |
| $G_i^c$ | The $i$-th trait cluster for class $c$ |
| $K$ | Number of independent trait extractions per image |
| $k_1$ | Number of traits retained after global filtering |
| $k_2$ | Number of traits retained after local refinement |
| $P$ | Number of patch tokens |
| $v_i$ | Visual embedding of patch $i$ |
| $\mathbf{v}$ | Global visual embedding (mean-pooling) |
| $\mathbf{e}_t$ | Embeddings of trait $t$ |
| $\mathbf{E}_\mathcal{T}$ | Trait embedding matrix |
| $d$ | Embedding dimension |
| $\mathbf{s}$ | Similarity score |
| $p_S$ | Prompt for semantic traits |
| $p_L$ | Prompt for low-level traits |
| $\mathcal{D}$ | Few-shot support set |
| $c$ | Class index |
| $C$ | Total number of classes |
| $n$ | Number of instance per class in support set |
| $x_i$ | Image instance $i$ |
| $y_i$ | Class label of instance $i$ |
| $x_{\text{test}}$ | Test image |

## B SELECTING DISTANT-OOD

Table 3: Distant-OOD Selection Results. We classify a dataset as distant-OOD if the performance gain ($\Delta$) from few-shot SFT over the zero-shot baseline is less than 15%.

| Dataset | Zero-shot (%) | 8-shot SFT (%) | $\Delta$ (%) | Distant-OOD? |
|---|---|---|---|---|
| *Standard Benchmarks (Not Selected)* | | | | |
| FGVC Aircraft | 25.31 | 51.82 | +26.51 | No |
| Flowers102 | 62.15 | 88.43 | +26.28 | No |
| Pets37 | 71.40 | 92.11 | +20.71 | No |
| Cars196 | 43.88 | 65.24 | +21.36 | No |
| *Distant-OOD Benchmarks (Selected)* | | | | |
| Pokemon | 48.92 | 52.85 | +3.93 | Yes |
| Retinal OCT | 18.75 | 33.39 | +14.64 | Yes |
| WM-811K | 11.45 | 15.90 | +4.45 | Yes |
| MVTec AD[*] | 34.88 | 36.71 | +1.83 | Yes |

[*]Result based on 1-shot SFT as per the dataset's experimental protocol.

As introduced in Section 4.2.1, a **distant out-of-distribution (distant-OOD)** task is defined as one where standard few-shot supervised fine-tuning (SFT) yields minimal performance gains over zero-shot baselines. This definition enables identification of challenging benchmarks where models cannot efficiently adapt through traditional parameter updates. The selection criterion is based on the performance gain ($\Delta$) from few-shot SFT over zero-shot baselines. A dataset is classified as distant-OOD if this gain falls below a $15\%$ threshold:

$$\Delta = \text{Acc}_{\text{few-shot SFT}} - \text{Acc}_{\text{zero-shot}} < 15\% \tag{3}$$

For most datasets, 8-shot SFT performance is used for this calculation. For datasets with limited samples per class, such as MVTec AD, 1-shot performance is employed. Twelve public datasets across various domains were systematically evaluated to identify those meeting this criterion. The detailed performance metrics used for selection are shown in Table 3. This empirical characterization establishes the foundation for developing efficient trait-based reasoning methods that can construct new visual knowledge without requiring parameter adaptation, addressing the computational and representational challenges inherent in distant-OOD scenarios.

## C   MVTec AD Detailed Results.

MVTec AD comprises 15 object/texture categories for industrial anomaly detection, where training sets contain exclusively normal samples and test sets include various defect types. The benchmark exhibits substantial variance in SFT performance across categories, with some categories demonstrating significant improvements while others experience substantial degradation. This heterogeneous behavior makes MVTec AD particularly suitable for efficiency-focused evaluation, as computational resources can be allocated more effectively to categories where adaptation proves beneficial.

Given the limited availability of defective samples, a 1-shot experimental setup is adopted to maximize computational efficiency: for each category, one defective image is sampled as support and evaluation is performed on the remaining test set. Each category is treated independently for defect classification to ensure fair comparison. Following the criterion established in Section 4.2.1, categories where SFT yields less than 15% performance change compared to baseline are selected for distant-OOD experiments, as these represent scenarios where traditional parameter adaptation methods demonstrate computational inefficiency without corresponding performance gains.

Table 4: MVTec AD Distant-OOD Selection. Categories are selected if the 1-shot SFT performance gain ($\Delta$) is less than 15%.

| Class | Zero-shot (%) | 1-shot SFT (%) | $\Delta$ (%) | Distant-OOD? |
|---|---|---|---|---|
| bottle | 51.90 | 48.10 | -3.80 | ✓ |
| cable | 9.22 | 41.13 | +31.91 | ✗ |
| capsule | 42.06 | 28.57 | -13.49 | ✓ |
| carpet | 56.76 | 54.95 | -1.81 | ✓ |
| grid | 52.78 | 40.28 | -12.50 | ✓ |
| hazelnut | 59.05 | 77.14 | +18.09 | ✗ |
| leather | 38.98 | 60.17 | +21.19 | ✗ |
| metal nut | 37.27 | 20.91 | -16.36 | ✓ |
| pill | 11.95 | 20.75 | +8.80 | ✓ |
| screw | 23.38 | 23.38 | 0.00 | ✓ |
| tile | 46.85 | 46.85 | 0.00 | ✓ |
| toothbrush | 55.00 | 57.50 | +2.50 | ✓ |
| transistor | 13.68 | 13.68 | 0.00 | ✓ |
| wood | 42.47 | 42.47 | 0.00 | ✓ |
| zipper | 28.67 | 27.97 | -0.70 | ✓ |

## D   Reproducibility

### D.1   Traits Extraction Prompts

```
enhanced_hybrid_categorized_prompt = (
    "You are an expert {dataset} {noun} observer. This is {article}
    {type_name} image. Focus ONLY on the {type_name}.\n\n"
    "Carefully analyze this {type_name} image and extract 8-15 visual
```

Table 5: Detailed 1-shot Results on Selected MVTec AD Distant-OOD Categories.

| Class | Zero-shot | SFT | SFT+LoRA | RFT | IVL |
|---|---|---|---|---|---|
| bottle | 51.90 | 48.10 | 53.16 | 53.16 | 54.21 |
| capsule | 42.06 | 28.57 | 40.48 | 42.06 | 45.13 |
| carpet | 56.76 | 54.95 | 58.56 | 54.95 | 59.32 |
| grid | 52.78 | 40.28 | 52.78 | 52.78 | 54.88 |
| metal nut | 37.27 | 20.91 | 36.36 | 20.91 | 39.75 |
| pill | 11.95 | 20.75 | 12.58 | 20.75 | 25.41 |
| screw | 23.38 | 23.38 | 23.38 | 18.83 | 26.92 |
| tile | 46.85 | 46.85 | 47.75 | 49.55 | 50.15 |
| toothbrush | 55.00 | 57.50 | 55.00 | 27.50 | 58.10 |
| transistor | 13.68 | 13.68 | 13.68 | 55.79 | 15.23 |
| wood | 42.47 | 42.47 | 41.10 | 58.90 | 45.66 |
| zipper | 28.67 | 27.97 | 27.97 | 17.48 | 31.89 |
| **Average** | **38.57** | **35.45** | **38.57** | **39.40** | **42.22** |

```
    traits that help classify the {noun} of this {type_name}.\n\n"
    "Categorize each trait into SEMANTIC (domain-specific) or
    LOW-LEVEL (universal patterns):\n\n"
    "SEMANTIC TRAITS:\n"
    "- [Domain-specific features like 'has fins', 'metallic body',
    'digital display']\n\n" "LOW-LEVEL TRAITS:\n"
    "- [Universal patterns like 'curved edges', 'smooth texture',
    'parallel lines']\n\n" "Guidelines for categorization:\n"
    "• SEMANTIC: Features specific to {type_name}s (body parts, materials,
    functions)\n"
    "• LOW-LEVEL: Universal visual patterns (shapes, textures, edge types,
    spatial relationships)\n\n" "RULES:\n"
    "- Use visible evidence only; do not guess or infer from world
    knowledge\n"
    "- Describe only the {type_name}, not
    background/props/text/watermarks\n"
    "- Order traits from most distinctive to least within each category\n"
    "- Each trait must be 2-6 words and describe what is actually visible
    in this {type_name}\n"
    "- No uncertainty terms (maybe/probably/seems), no punctuation
    except hyphens\n\n"
    "Output format (use exact section headers):\n\n"
    "SEMANTIC TRAITS:\n"
    "- <semantic trait 1>\n"
    "- <semantic trait 2>\n"
    "- <semantic trait 3>\n\n"
    "LOW-LEVEL TRAITS:\n"
    "- <low-level trait 1>\n"
    "- <low-level trait 2>\n"
    "- <low-level trait 3>"
)
```

## D.2 BASELINE

Our experimental configurations were designed to ensure computational efficiency while maintaining methodological rigor across all baseline comparisons. The 7B Qwen2.5-VL model served as the foundation for zero-shot inference across all experiments to establish computational consistency and enable fair comparison. For parameter-efficient adaptation methods, supervised fine-tuning (SFT) and SFT with LoRA configurations were implemented using the same 7B Qwen2.5-VL backbone through the LLaMA-Factory framework, which provides standardized and computationally efficient implementations. Training procedures followed consistent protocols with 8 epochs and default hyperparameter settings to maintain reproducibility, while random seeds were employed for initialization consistency. Similarly, reinforcement fine-tuning (RFT) experiments utilized the identical model backbone through the Visual-RFT pipeline (Liu et al., 2025), with all default configuration settings preserved to ensure experimental consistency. Random seed initialization protocols matched those used for SFT implementations to maintain experimental validity. This unified experimental framework ensures that observed performance differences reflect genuine methodological advantages rather than implementation artifacts or architectural variations. The consistent computational configuration across all methods enables di-

Table 6: Type and Unit Names for Each Dataset

| Dataset | Type Name | Unit Name |
|---|---|---|
| FGVC Aircraft | aircraft | model |
| Flowers102 | flower | species |
| Pets37 | pet | breed |
| Cars196 | car | model |
| Pokemon | Pokemon | type |
| WM811k | wafer map | failure type |
| Retinal OCT | retinal | condition |
| MVTec AD | (sub-category) | defect |

rect efficiency comparisons and validates the computational benefits of the proposed trait-based approach over traditional parameter adaptation strategies.

```
This is {article} {type name} image.
Please classify this {type name} image into one of the following
categories: {list of all class names}.
Output the thinking process in <think> </think> and final answer in
<answer> </answer> tags.
The output answer format should be as follows:
<think> ... </think> <answer>{unit name} name</answer>
Please strictly follow the format.
```

Type names and unit names for each dataset are specified in Table 6. Article is set to either 'a' or 'an'.

### D.3 IN-CONTEXT LEARNING (ICL) EVALUATION

Few-shot ICL was evaluated without parameter updates to assess computational efficiency. For each query image, $k \in \{1, 8\}$ demonstrations (image + instruction + gold answer) were prepended, drawn uniformly at random from the same dataset (excluding the query) using a fixed seed (42) and canonical sort to stabilize ordering. The query block contained only the image and instruction. Decoding employed temperature $= 0.0$, top-$p = 1.0$, and max_new_tokens $= 500$. Predicted classes were extracted from the first <answer>...</answer> span, with labels normalized (lowercase, trimmed punctuation) before exact-match scoring. Accuracy per shot was reported with JSONL logs containing seeds, selected demonstration IDs, token counts, raw generations, parsed answers, and correctness to enable full reproduction.

#### D.3.1 ICL PROMPTS USED.

**Conversation Content**

```
This is an example of {CLASS TYPE}.

<image: {demo_i.jpg}>

Valid categories (choose exactly one):\n{bullet_list}\n\n
Classify the following test image using only the exact labels from this
list.

<image: {test_image.jpg}>

Output the thinking process in <think> </think> and final answer in
<answer> </answer> tags. The <answer> tag must contain exactly one label
copied verbatim from the list above.
Format example: <think> ... </think> <answer>Example Label</answer>\n
Do not include any text outside the tags.
```

## D.4 DATASET

### D.4.1 PUBLIC DATASETS

For most existing datasets, 8 examples were randomly sampled from each class to form the 8-shot training set, with further subsampling for the 1-shot training set, while the remaining examples constituted the test set. For MVTec AD dataset, which contains 15 defect detection categories, experiments were conducted on each category individually. Due to dataset size limitations, only 1-shot training sets were sampled to ensure computational feasibility while maintaining experimental validity.

### D.4.2 POKEMON DATASET

To establish a controlled distant-OOD benchmark, images of 1,025 Pokémon creatures were collected from the Pokémon Database (Database, 2025; Nintendo, 2025) to ensure high visual quality and comprehensive coverage. Approximately half of the Pokémon possess dual-type classifications, requiring modified evaluation criteria: since the experimental prompt instructed models to select a single type, either correct type was accepted as a valid prediction for dual-type specimens.

Human performance baselines were established through systematic evaluation across three expertise levels, corresponding to different model training states. Beginners, familiar only with prominent Pokémon such as Pikachu, served as comparisons to base models like 7B Qwen2.5-VL. Intermediates, possessing knowledge of Pokémon type associations including color schemes and morphological patterns, represented human equivalents to few-shot trained models. Experts, having completed Pokémon games or possessing extensive familiarity, corresponded to fully trained model performance.

Three questionnaire versions were constructed, each containing 341-342 Pokémon to categorize, distributing the complete dataset evenly. Each version maintained similar distributions of well-known Pokémon across all expertise levels to ensure balanced evaluation. For dual-type evaluation, participants could select 1-2 types per Pokémon, with strict criteria: single-type Pokémon required exact matches, while dual-type Pokémon required both types to be correctly identified for credit.

During a one-week collection period, 45 responses were gathered across 15 beginners, 16 intermediates, and 14 experts. This human baseline establishes computational efficiency benchmarks by demonstrating that tasks trivial for humans through brief exposure remain computationally intensive for VLMs despite extensive parameter adaptation.

Table 7: Human Evaluation on Pokemon Dataset

| Expertise Level | Accuracy |
| --- | --- |
| Beginner | 55.4% |
| Intermediate | 61.9% |
| Expert | 84.3% |

## E LLM USAGE

We used LLMs to polish our writing, including grammar checking, rephrasing, and organizing the flow. We also used LLMs for searching related work that might be relevant to our research. Additionally, we discussed our ideas with LLMs to improve our initial thoughts.

