# OpenReview forum: "Inductive Visual Logic for Few-Shot Out-of-Distribution Adaptation in VLMs"
_ICLR.cc/2026/Conference — ICLR 2026 Conference Withdrawn Submission_

### Official Review · Reviewer_dDUm · 2025-10-25

**Soundness:** 2
**Presentation:** 2
**Contribution:** 3
**Rating:** 4
**Confidence:** 4

**Summary:**

This paper introduces Inductive Visual Logic (IVL), a training-free framework for few-shot distant out-of-distribution adaptation in vision-language models (VLMs).
Instead of parameter updates or prompt tuning, IVL constructs an explicit trait-based reasoning pipeline that mimics human inductive learning:
(1) extract both semantic and low-level visual traits from support examples via dual-mode prompting;
(2) cluster and canonicalize traits into interpretable descriptors;
(3) build a trait database; and
(4) at inference, perform global similarity retrieval, localized attention-based verification, and multi-modal reasoning using the original image and cropped evidence regions.
Experiments on four distant-OOD datasets show IVL consistently outperforms fine-tuning (SFT/LoRA), reinforcement fine-tuning (RFT), and in-context learning (ICL), despite requiring no gradient updates.
The paper argues that distant-OOD failures stem from the absence of representational primitives in pretrained features and that trait-level reasoning provides a scalable, interpretable alternative.

**Strengths:**

1. Clearly defines distant OOD as cases where fine-tuning yields <15% improvement, offering an operational criterion to distinguish near vs. distant OOD.
2. Introduces a coherent, end-to-end trait-based reasoning framework combining semantic and low-level feature extraction, clustering, and attention grounded verification.
3. Fit human inductive reasoning (trait identification and abstraction) with computational implementation, forming an interesting cognitive–computational bridge.
4. Produces explicit visual traits and grounding regions, aligning model predictions with human understandable explanations.

**Weaknesses:**

1. The paper lacks analysis on individual component contributions (e.g., semantic vs. low-level traits, canonicalization, local verification). This omission weakens causal attribution of the gains.
2. The core mechanism of IVL relies on generating textual trait descriptions from the support set via the VLM and performing similarity-based matching within a shared vision–language embedding space. This remains a non-parametric retrieval process rather than introducing new learning operators or reasoning mechanisms. The “logic” component mainly serves as a linguistic interpretation layer rather than an algorithmic innovation, making the contribution more about system integration and reframing than a fundamentally new learning paradigm.

**Questions:**

1. How does performance change when removing or isolating key components ,  e.g., using only semantic traits, only low-level traits, skipping canonicalization, or disabling local attention refinement?
2.  Would IVL still work if applied to a different pretrained VLM (e.g., LLaVA-1.6 or BLIP-2)? How sensitive is it to the embedding space of Qwen2.5-VL? (I am mainly curious about this aspect and it would not affect my overall rating.)
3. Since IVL performs multiple trait text queries per test image, how does its inference cost compare to fine-tuning or ICL?
4. Since IVL is a non-parametric retrieval-based approach, the trait dictionary size and inference cost are expected to grow with the number of classes. It would be important to understand whether this scaling affects performance or efficiency in practice.

---

### Official Review · Reviewer_VYwN · 2025-10-27

**Soundness:** 2
**Presentation:** 2
**Contribution:** 2
**Rating:** 2
**Confidence:** 5

**Summary:**

The paper proposes Inductive Visual Logic (IVL), a training-free framework for adapting vision-language models to few-shot distant out-of-distribution settings where fine-tuning fails. IVL replaces parameter updates with explicit trait-based reasoning as claimed: it extracts semantic and low-level visual traits from a few examples, clusters and canonicalizes them into interpretable trait dictionaries, and performs hierarchical inference by retrieving and grounding relevant traits before final classification. This approach allows models to reason over observable visual evidence rather than rely on missing pretrained features. Experiments on Pokémon, Retinal OCT, WM811k, and MVTec-AD show that IVL improves accuracy and interpretability over supervised fine-tuning, LoRA, reinforcement fine-tuning, and in-context learning, while defining a clear benchmark for distant-OOD adaptation.

**Strengths:**

Strengths

**Originality:** The paper introduces a distant-OOD setting that isolates cases where existing VLM adaptation methods fail, and proposes the novel Inductive Visual Logic (IVL) framework that replaces gradient-based adaptation with explicit, interpretable trait-based reasoning inspired by human cognition.

**Quality:** The method section is well-structured, with clear stages (trait extraction, clustering, hierarchical inference), and an experimental design across multiple datasets, demonstrating robust improvements over strong baselines while remaining fully training-free.

**Clarity:** The paper is clearly written, with intuitive figures and step-by-step methodological explanations.

**Significance:** The paper demonstrates that reasoning over explicit visual traits offers a practical and interpretable alternative to fine-tuning, highlighting a promising direction for few-shot adaptation of large VLMs in high-stakes domains such as medical imaging and industrial inspection.

**Weaknesses:**

I will list the weakness of the paper in the following:
1. The paper lacks internal ablations on its key design choices, making it unclear which components of IVL are truly responsible for the observed gains.

2. No ablation separating semantic vs low-level trait extraction is provided, despite this being a central claim of the method’s robustness under domain shift.

3. The trait clustering and canonicalization process (Sentence-BERT vs VLM encoder, and the filtering strategy) is not analyzed, leaving uncertainty about its real impact on performance and interpretability.

4. The hierarchical inference pipeline (global retrieval, attention-based grounding, region cropping, final reasoning prompt) is not broken down experimentally; thus, the contribution of each stage to accuracy and interpretability is unverified.

5. The computational cost and scalability of maintaining large trait dictionaries (up to 10k traits) are not discussed, raising questions about efficiency for real-world deployment.

**Questions:**

I consider the weaknesses of the paper for all questions I have. I would appreciate the authors addressing them.

---

### Official Review · Reviewer_KZnv · 2025-10-29

**Soundness:** 1
**Presentation:** 3
**Contribution:** 3
**Rating:** 4
**Confidence:** 5

**Summary:**

The motivation of the paper is strong and addresses a critical failure of VLMS for real OOD (ditstant OOD) adaptation. The authors have clearly defined the reseach problem, and proposes a training free solution based on trait extraction and logical filtering. IVL's strength depends on iterpretability of selected VLM and its ability to leverage high-level semantic knowledge and low-level visual primitives.

However, lack of strong empirical results, especially ablation results on scalability, applicability and how individual components contribute to the final solution critically impacts the soundness of the paper. The paper makes an important technical contribution, but the present experimental support  does not completely support it.

**Strengths:**

- **Originality** : IVL is a training free algorithms completely bypasses the limitations of gradient based and in-context learning approaches for distant-OOD. The authors identifies and addresses a critical failure model of large VLMS and proposed a data efficient solution.

- **Algorithm**: Dua-mode trait extraction is an interesting design, ensuring tht system can still extract primitive feature even when the semantic features for a new domain is absetn.

- **Sturcture**: The paper is well written. From introducing the problem to proposing a viable solution, authors logically and intuitively connected the dots.

**Weaknesses:**

- **Empirical Evidences**: The experimental setup to support the claim is incomplete. Authors have provided the main task results for single large scale model, which questions the scalability and adaptability of proposed solution.

-**Interpretability**: Lack of evidences/expriments for the paper claim of 'more interpretable predictions". Most of the steps of the solution, textually elaborated, there's no evidences or examples to show what is the part doing and how does it contribute.

-**VLM's Capability**: The algorithm heavily depends on inherent performance of the base model in extracting traits and identifying visual patterns. Without a  pragmatic evaluation,  it becomes difficult to understand  how the component contributes.
- **Clustering** : The success of IVL heavily depends on trait clustering.  Can the authors discuss sensitivity of the final performance of HDBSCAN parameters and VLM's quality in generating canonical name for diverse trait clusters?. What are some alternative clustering methods can be used in this context, and how do they affect the performance of IVL?

**Questions:**

1.What is the performance gain from using Dual-mode Trait Extraction ($t^S+t^L$) compared to using only Semantic ($t^S$) or only Low-level ($t^L$) traits?
2.How robust is the Trait Building stage when the base VLM is changed (e.g., from Qwen-VL to LLaVA or MiniGPT-4)?
3. Does the generated set of canonical traits transfer effectively across different base models, or must the Trait Building be re-run for every VLM?
3. How much does the Localized Visual Grounding step (attention-based refinement and region cropping) contribute to the final accuracy compared to using only the Global Trait Retrieval ($\mathcal{T}^{(1)}$) for classification?
5. There are multiple LLM calls involved. What is the trait building cost?
6. What are the failure mode of IVL?
7. The architecture diagram can be improved, especially the train clustering and filtering. Use an ordering numbers and explain the steps in the description.

---

### Official Review · Reviewer_pbwH · 2025-11-01

**Soundness:** 2
**Presentation:** 3
**Contribution:** 2
**Rating:** 4
**Confidence:** 3

**Summary:**

The paper tackles the problem of adapting vision-language models in a few shot manner to distant OOD tasks. The main motivation for the approach stems from the observation that fine-tuning internal representations and in-context learning struggle for downstream tasks that are far from pre-training data distribution. The core idea is to construct trait dictionary offline via dual prompting strategy aimed at capturing both semantic-level traits and low-level traits. At inference time, hierarchical trait filtering and localized visual grounding is performed for a given test image to decide the category label. Results claim to achieve better performance for 1 and 8 shot cases for a single VLM baseline on different OOD datasets.

**Strengths:**

- The problem of producing good classification performance for far OOD downstream tasks is relevant and challenging. There are many narrow downstream tasks where data distributions are far away from the pre-training data for VLMs.

- The idea of maintaining explicit traits at semantic and low-level is interesting where it is used to produce different kinds of evidences for final classification with the VLMs.

- The motivation for the idea is well-described in the introduction.

- Experiments have been performed on different OOD datasets to validate the effectiveness of the method and the results claim to obtain superior performance than competing methods.

**Weaknesses:**

- Experiments have been performed with only single VLM baseline model. It is not clear if the method can work with other VLMs?

- Table 1: the gains in 2/4 datasets seems marginal over SFT baseline.

- The paper claims to be proposing an efficient method at various places, however, there is no computational analyses comparison with other competing approaches.

- The paper lacks ablation studies at all. It is not clear how
    - each of the main components,
   - design choices and
   - selection of hyperparamters
   - contribute to the overall performance gains.


- The methods section is sometimes hard to follow, for example 3.4.2 and 3.4.3 where references to figures containing illustrations are very important for proper understanding.

**Questions:**

- L298: how relevance filtering is performed to discard inconsistent traits?

---

### Note · Authors · 2025-11-12

**Comment:**

We sincerely thank the reviewers for their insightful feedback and for recognizing the originality of our method. We are withdrawing to strengthen the work with comprehensive experiments and analyses.

**Withdrawal Confirmation:**

I have read and agree with the venue's withdrawal policy on behalf of myself and my co-authors.